# Linear breakwater reefs of the greater Caribbean: Classification, distribution & morphology

Paul Blanchon[1]*, Alexis E. Medina-Valmaseda[2], Eduardo Islas-Domínguez[2], Edlin Guerra-Castro[3], David Blakeway[4], Joaquín Rodrigo Garza Pérez[5], Adan Guillermo Jordan-Garza[6], Ismael Mariño-Tapia[7], Paula A. Zapata-Ramírez[8]

1 Reef Systems Unit (Puerto Morelos), Instituto de Ciencias del Mar y Limnología, Universidad Nacional Autónoma de México, Puerto Morelos, Q. Roo, México, 2 Posgrado en Ciencias del Mar y Limnología, Universidad Nacional Autónoma de México, Coyoacán, Ciudad de México, México, 3 Escuela Nacional de Estudios Superiores Unidad Mérida, Universidad Nacional Autónoma de México, Mérida, Yucatán, México, 4 Fathom 5 Marine Research, Lathlain, Western Australia, Australia, 5 Unidad Multidisciplinaria de Docencia e Investigación-Sisal, Facultad de Ciencias, Universidad Nacional Autónoma de México, Sisal, Yucatán, México, 6 Coral Reefs Laboratory, Facultad de Ciencias Biológicas y Agropecuarias, Cuerpo Académico Ecosistemas Costeros, Universidad Veracruzana, Tuxpan, Veracruz, México, 7 Departamento de Matemáticas Aplicadas y Computación, Escuela Nacional de Estudios Superiores Unidad Mérida, Universidad Nacional Autónoma de México, Mérida, Yucatán, México, 8 Escuela de Ingeniería, Grupo de Automática y Diseño A + D, Universidad Pontificia Bolivariana, Medellín, Colombia

* blanchons@gmail.com

**Data Availability Statement:** All relevant data are within the paper and its Supporting Information files.

## Abstract

Geomorphic differences among Caribbean reefs have long been noted. These differences are considered to reflect the presence of reefs in different stages of development, following an incomplete recovery from rapid deglacial sea-level rise. But the possibility that these reflect real developmental differences caused by variation in wind, wave, and climate regime, has never been fully considered. Here, for the first time, we quantify the geomorphology and distribution of Greater Caribbean reefs using satellite images in Google Earth and public-domain bathymetry. To do this, we first standardise their classification based on shallow geomorphology, substrate depth, and physiographic setting, and then count and categorise the total number of reefs. These data show a total of 1023 linear breakwater reefs with a combined length of 2237 km. Of this total length, 80% are fringing reefs, 16% are barriers and 4% are faros and atolls. In terms of categories, there are 16 reef subtypes present, but only 9 are common. Their distribution, however, is not uniform. In particular, flat-subtypes form 60% of breakwater reefs in southern regions, but are less common in northern regions where crest-subtypes dominate (80%). To distinguish the geomorphology of these common reef subtypes, we collect size- and length-related morphometric data from their main reef zones. These data reveal that flat and crest subtypes also have morphometric differences: flat subtypes have well-constrained morphologies with statistically consistent unimodal morphometrics characterised by large back-reef zones, smaller and steeper reef fronts, and more sinuous and persistent crestlines. Crest subtypes, by contrast, have multimodal morphometrics suggesting less consistent morphologies (or unresolved subtypes), and are characterised by crestlines with lower sinuosity, more variable back-reef and reef-

**Funding:** This work is supported by SEP-CONACyT grant A1-S-18879 and PAPIIT grant IN214819 to PB. The funders had no role in study design, data collection and analysis, decision to publish, or preparation of the manuscript.

**Competing interests:** The authors have declared that no competing interests exist.

front areas, and slopes. These differences in geomorphology and distribution imply that flat- and crest-subtypes are neither successional stages of a single reef type, nor a genetically related sequence of types, but distinct reefal geoforms with different modes of development. In subsequent work we will explore what controls these differences.

## Introduction

"...the principal kinds of coral-reefs...were found to differ little, as far as relates to the actual surface of the reef. An atoll differs from an encircling barrier-reef only in the absence of land within its central expanse; and a barrier-reef differs from a fringing-reef, in being placed at a much greater distance from the land with reference to the probable inclination of its submarine foundation, and in the presence of a deep-water lagoon-like space or moat within the reef." (Darwin p.146, 1842)

By recognising their superficial similarity, Darwin [1] postulated that the three principle reefs were actually a single type with three stages of developmental: in the first stage, fringing reefs formed around volcanic islands, and then in subsequent stages developed vertically into barrier reefs and atolls as the islands subsided. But having to confirm subsidence in order to distinguish these stages subsequently led to many difficulties in their classification. These were particularly acute in the Caribbean where, in contrast to the Indo-Pacific, Darwin found reefs to be discontinuous linear tracts that, inexplicably, lay some distance back from the shelf edge. Even in Belize, where they appeared to be more classic barriers and atolls, reefs still had significant sections charted as sand banks devoid of coral. Similarly, out in the Atlantic, Bermuda appeared to be an atoll, but had a discontinuous reef rim, an anomalously wide shallow shelf, and high islands of windblown sand. These enigmatic differences consequently forced Darwin to leave many Caribbean reefs unclassified.

Differences in the geomorphology of Caribbean reefs continued to pose problems when underwater scuba exploration began in the mid 1950´s. Echoing Darwin's comparison, initial work in the Bahamas reported that reefs developed back from the bank edge, lacked intertidal reef flats, and only partially covered the bedrock foundation, suggesting they were underdeveloped when compared to Indo-Pacific reefs [2, 3]. However, this immaturity contrasted with reports of better developed fringing reefs along the north coast of Jamaica, which were closer to the shelf edge, and fully covered the narrow shelf with diverse multi-zoned communities of corals [4]. Although these 'climax' reefs were certainly more comparable with their Indo-Pacific cousins, they nevertheless lacked extensive reef flats and were discontinuous along shore. Furthermore, they were replaced along Jamaica's south coast by small mid-shelf barrier-reefs that were almost barren of corals, and thus interpreted to be in a regressed ecological state due to storm exposure [4, 5]. These morphological contrasts between reefs in the Caribbean continued to divide opinion throughout the 1970s, with some claiming that they were comparable to Indo-Pacific reefs, having inter-tidal reef flats and diverse communities that extended to 70 m of water [6, 7], and others that they had limited accretion, formed thin layers over residual topography, and were thus underdeveloped due to a lack of subsidence [8–11].

Adding to the controversy, significant differences were also found when portable drills were first used to investigate reef development in the mid 1970s. These drill data quickly showed that reefs were not merely veneers over Pleistocene topography as had been widely postulated, but instead had developed as much as 33-m-thick sequences in the last 8 ka [7, 12,

13]. Some of these sequences were built entirely by reef-crest corals, suggesting vertical growth matched sea-level rise [7, 12]. But others had thinner coral caps underlain by extensive units of skeletal sediment [13–15]. This sediment was thought to accumulate in the mid shelf and facilitate reef catch-up, following an initial stage of reef suppression due to suboptimal water quality during early Holocene shelf inundation [14]. Such differences in internal structure were thus attributed to local variations in shelf morphology [16].

An explanation of why Caribbean reefs show differences in both surface geomorphology and internal structure has never been fully addressed, but early arguments implied that they were in various ecological stages of development [17, 18]. For example, Goreau [7] stated that "...*modern reefs are not stable and mature communities but are undergoing successional changes typical of youthful assemblages*" (p. 323). He continued that their intermittent lateral development and assemblage changes were due to their youth and insufficient time to eliminate random effects in the 5 ka since sea level had stabilised. As Meischner and Meischner [19] succinctly put it: SL rise was too fast for morphological equilibrium and the Holocene was too short for faunal recovery. Similar explanations were also used to explain reef differences outside the Caribbean. For example, Hopley [20] proposed that reef subtypes on the Great Barrier Reef represented distinct stages of vertical and lateral growth from topographic residuals, including the amalgamation of patch reefs to form 'crescent' reefs, their lateral extension to form 'lagoon' reefs, and a final infilling stage producing 'planar' reefs. He considered that emergent reefs were absent during rising sea level and their 'maturity' was thus controlled by the initial amount of vertical growth required to reach sea level.

Such successional ideas follow Darwin's basic tenet that reefs are genetically related in space and time, yet ignore the possibility that there may be more than one type of coral reef. If we consider that spatio-temporal variation in reef ecology shapes geomorphology, which in time extends into geological development, then reef types should develop differently under different environmental conditions [21]. To test this hypothesis we open an investigation into the geomorphology of breakwater reefs in the Greater Caribbean province using Google Earth, which provides high resolution and extensive image coverage. Our sole objective in this paper is to document the abundance and distribution of reef types and collect morphometric data that allows their morphologies to be differentiated. To accomplish this objective, we first standardise the classification of reef types so that we can count and categorise the total number of linear reefs present in the Greater Caribbean and document their geographic distribution. We then collect morphometric data from a subset of these reef types to calculate their degree of internal and external morphological variance and identify morphotypes. In subsequent papers we will further analyse these data using space-for-time substitution to explore the principal factors controlling the geomorphology, distribution and development of common reef types.

## Terminology and methods

### Reef categories

In order to count all breakwater reefs in the province, we first systematically categorise reef geomorphology by amplifying and standardising Darwin's canonical categories of coralgal reefs using empirical criteria, rather than genetic interpretation. As shown in Fig 1, we subdivide reefs at 4 levels using depth (Intertidal or Submerged), form (Linear or Nonlinear), physiographic setting (Interior, Coastal, Bank and Oceanic) and shallow geomorphology (Flat-type or Crest-type). These levels provide objective and widely applicable criteria by which to differentiate all reef types.

We define *Coralgal Reefs* as three-dimensional bio-accretionary structures (or geoforms) that actively produce the substrate upon which they grow [6]. As highlighted in Fig 1, those at

**Fig 1. Standardised geomorphic classification of coral reefs showing the subcategories of linear breakwater reefs.** There are 4 levels: The first is based on depth, with two categories, intertidal *breakwater* reefs, and subtidal *submerged* reefs. The second is based on form, with two categories, *linear* BW reefs and non-linear or *dispersed* BW reefs. The third, for linear BW reefs, is based on physiographic setting with four categories, *Interior* (protected Lagoons and Bays), *Coastal* (continental or insular shelves ≤ 5 km wide), *Bank* (waters with a shelving seafloor shallower than 200 m, including shelves >5 km wide) and *Oceanic* (waters lacking a shelving seafloor shallower than 200 m). The fourth is based on shallow geomorphology with two categories, *Crest-type* (sloping to subtidal depths on either side of the crest) and *Flat-type* (with an intertidal platform behind the crest). A final modification term for Fringing reefs is *Attached* to the coast (with no lagoon substrates, only back-reef) or *Detached* from it (with a lagoon). In total, this produces 32 possible categories of linear BW reef.

the sea-surface are termed *breakwater reefs*, and are defined as reefal geoforms that impede the passage of fairweather waves. Linear breakwater (BW) reefs form narrow elongate tracts with distinctive parallel benthic zones. There are four primary linear BW reef types: *fringing*, *barrier*, *faro*, and *atoll* (Fig 1). Fringing reefs are defined as linear low-relief tracts that rise <10 m above their surrounding substrate, whereas barrier reefs are linear tracts that rise 10 m or more above their substrate. Faros and atolls are defined as annular reef tracts that almost completely enclose a landless body of water (75% or more), and are distinguished based on size (not depth), with faro's having a diameter of <5 km and atolls 5 km or more. We ignore genetic use of the term atoll (resulting from subsidence), preferring it to be defined more simply from the annular form of the reef. Also, for the time being, we consider neither dispersed breakwaters nor submerged reefs (Fig 1).

The 4 linear BW reef types defined above are subdivided into 4 physiographic settings: lagoons and bays (1), coastal shelves (2), epeiric shelves and banks (3), and open seas and oceans (4). Linear BW reefs that develop in wave-protected lagoons and bays use the prefix *interior* (eg. interior fringing reef). Those on narrow shelves ≤5 km wide use the prefix *coastal* (eg. coastal fringing reef), whereas those on epeiric shelves or banks wider than 5 km use the prefix *bank* (eg. Bank fringing reef). And the prefix *oceanic* is used for linear BW reefs developed in open seas or oceans with no shelving substrate and water depths greater than 200 m (eg. oceanic atoll). This combination of 4 reef types and 4 physiographic settings produces 14 reef categories, not 16, since by definition fringing reefs cannot exist in oceanic settings and atolls (with diameters of >5 km) cannot fit on coastal shelves (with widths of <5 km).

We then further subdivide these 14 reef categories based on shallow reef geomorphology: *crest-type* reefs are those with a topographic crest, where the back-reef and reef-front substrates gradually slope away on either side of the crestline to subtidal depths; whereas *flat-type* reefs are those with an intertidal reef-flat, where the back-reef zone is a sub-horizontal intertidal platform. Adding this geomorphic subdivision thus produces 28 categories. A final modifier is applied based upon the relation of fringing reefs to the shore in both interior and coastal settings, and can be either *attached* (without a lagoon) or *detached* (with a lagoon). Barrier reefs, which must have a deep lagoon by definition, are always detached from shore. This therefore adds a further 4 subdivisions, giving a total of 32 potential categories of linear BW reef (Fig 1).

## Geomorphic zones and morphometrics

Each of the 4 linear BW reef types is composed of a coralgal sedimentary deposit that, due to vertical and lateral accretion, develops a three-dimensional topographic relief above the sea floor (i.e., a geoform). However, coral growth is not always restricted to these self-generated geoforms, and can also be found colonising adjacent rocky substrates and slope breaks, thus forming a wider ecological 'seascape' [17, 22]. Coral communities that merely veneer these adjacent substrates are termed *coral grounds*, and when covering residual bedrock topography produce *pseudo reefs* (which can also develop over man-made objects). Such ecological units are not accreting reef geoforms, and so are not considered further.

As shown schematically in Fig 2, the coralgal sedimentary deposit of linear BW reefs consists of 2 main geomorphic units: a *back-reef* (BR) unit and a *reef-front* (RF) unit, separated by a *crestline* (CL) where waves break. (These geomorphic features and zones are defined in the S1 File). In well-developed reefs, these units are contiguous and only become non-contiguous where reef development is incomplete. We use breaks in the reef deposit—through both back-reef and reef-front units—to separate individual reefs. The 2 main geomorphic units that form the reef deposit can be divided into subzones based on ecological, benthic or geomorphic character [12, 23, 24].

To document the abundance and distribution of BW reefs in the Greater Caribbean, we count all visible geomorphic structures longer than 500 m but, to ensure full development, only collect morphometric data from those $\geq$ 1 km in length (Fig 2). To facilitate a direct comparison of their geomorphic structure, we sample a representative 1 km-long section from each reef where the crestline and boundaries of the back-reef and reef-front deposit can be clearly delineated from Google Earth imagery (Fig 2). From this sample, 4 metrics are calculated: First, two parallel-sided polygons joined back-to-back at the crestline are delineated, one representing the reef-front area and one the back-reef area (Fig 2). Second, the surface areas of these polygons are summed to calculate the total area of the reef deposit, and derive a scale-independent ratio of their relative size. In addition, a further 6 morphometric parameters are derived for each reef sampled. Two metrics are derived from the persistence of the crestline, which is the net distance over which it forms a breakwater and is determined by summing gaps in its total length (not sampled length) and calculating them as a percentage. Gaps are identified by breaks in the crestline (but not in the reef-front or back-reef zones), lack of wave breaking in retrospective images, and wave diffraction into the lagoon. Two metrics are also derived from the crestline sinuosity, which is the degree to which it deviates from a linear form and results from the difference between the total crestline length of the entire reef (not the 1 km sample) and its direct end-to-end distance. In very long reefs, sinuosity can be affected by coastal curvature and so, in these rare cases, the reef is subdivided to exclude strongly curved segments.

To define the main reef types, the depth of the lagoon and fore-reef shelf on either side of the reef deposit is determined directly from public-domain bathymetric data (Navigation

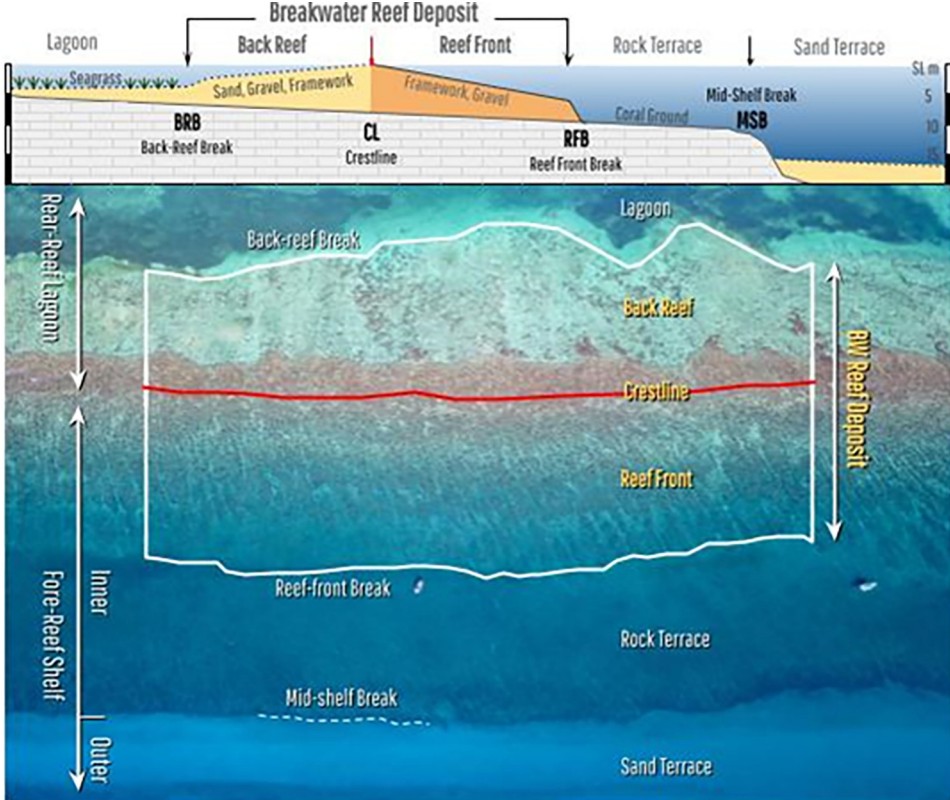

**Fig 2. Breakwater reef terminology and geomorphic zonation of shallow benthic seascapes in the Greater Caribbean.** The upper profile shows that BW reefs are three-dimensional bio-accretionary benthic structures (geoforms) composed of two geomorphic units—the back-reef and the reef-front—either side of the crestline, where fair-weather waves break. These two units constitute the BW reef deposit. As well as covering the reef deposit, corals are also found veneering non-reefal benthic zones in the wider ecological seascape, such as in the adjacent lagoon and inner fore-reef shelf. The outer fore-reef shelf can also be a depocentre for sand and submerged reef development. The lower image illustrates how a 1-km long section of each reef is sampled to determine 10 morphometric parameters: polygons delineate back-reef and reef front areas, which combine to give total deposit area and size ratio; actual total crestline length is compared to end-to-end length to derive sinuosity, and its persistence is derived from summing the gaps as a percentage of the total length; the reef-front slope is derived from the average distance from the crestline to the mid-shelf break (or 15 m isobath); These 10 morphometric parameters are used to statistically analyse the geomorphology of reef types. Drone image courtesy of Dr. L. Alvarez-Filip.

Charts, National Surveys, published literature etc). In the absence of bathymetric data, proxy indicators of depth and slope are used. Shelf and lagoon depths exceeding 10 m, for example, show a distinct texture or tone contrast between sandy substrates and can be corroborated by the draft of large shipping vessels (or corroborated when data becomes available). A proxy for the reef-front slope is determined by measuring the distance between the crestline and the *Mid-Shelf Break (MSB)*, which is a widespread slope break delimiting the edge of the inner fore-reef shelf and occurs between the 10–15 m isobath (Fig 2; [24]). The MSB position is identified from the texture/tone contrast between the shallow rocky fore-reef terrace and the lighter-toned deep sand terrace (Fig 2). This contrast may be enhanced by the growth of corals on the shoulder of the slope break and, in some areas, significant submerged reef development (e.g. [25, 26]). In areas of extensive reef-front development, distinguishing the MSB can be difficult but is approximated using the tone contrast created by sediment deposition below the 15 m isobath. Two metrics are derived to characterise the reef-front slope: the distance between the crestline to the MSB and its gradient, derived by dividing the rise at the MSB (15 m) over

the run distance to the crestline (i.e. Rise 15 m/Run distance x 100 = % slope). The gradient produced is categorised as low <3% (flat shelf), medium 3–10% (inclined shelf), and high >10% (steep shelf).

## Statistical analysis

In addition to counting and categorising reefs, we also make a preliminary analysis of the morphological variation within and between reef types by collecting independent morphometric data consisting of linear, ratio and percentage variables from the most common types (Fig 2). In the first step of this analysis, we delineate the degree of internal morphological variation within categories in order to identify which types have distinct morphologies and which don't. This is achieved by calculating averages of each morphometric parameter, and their corresponding variance expressed as average deviations from the centroid. To visualise morphometric variation within each reef type, kernel density estimates are used to identify distinctive metrics. Combined morphometric variation is then compared between reef types using a distance-based multivariate analysis performed using a dissimilarity matrix of Euclidean distance derived from the normalised morphometric variables (centred on zero in a common scale). To test the hypothesis of equal morphometric variance between reef types, we conduct a pairwise comparison of mean Euclidean distances to group centroids using a permutational analysis of within-group multivariate-dispersion, which is equivalent to Levene's test for equality of variances (PERMDISP, [27]). The central tendency of data from each reef type within the multivariate space is represented using bootstrap average regions [27].

In the second step, we identify which morphometrics confer differences between reef types. To do this we use not only the averaged metrics but also the relative contribution of each morphometric variable to each group's internal variability (i.e., consistency) identified by breaking down the similarity percentage within each group using the SIMPER routine [28]. Differences at the main BW classification levels (main type, physiographic setting, and shallow geomorphology) are quantified using a multivariate Discriminant Canonical Analysis of Principal Coordinates [29, 30]. The strength of the association between the multivariate morphometric properties and the classification levels are indicated with canonical correlations, and tested using a cross-validation procedure (leave-one-out, assigning observations to groups) for each level. All multivariate analyses use PRIMER v7 & PERMANOVA (PRIMER-e, Quest Research Limited [31]). Graphical outputs use open libraries available in R-CRAN [32].

## Results

### Reef length and abundance

As shown in Table 1, there are a total of 1023 linear BW reefs >500 m in length in the Greater Caribbean, with a total length of 2,237 km, and a mean of 2.2 km (all reef data are included as kmz files in S1 File). The longest unbroken reef tract, at 34 km, is the barrier reef at Alacranes in the Gulf of Mexico, closely followed by the 33 km long fringing Lighthouse Reef in Belize, with the fringing reef around Chinchorro Bank coming in 3rd place at 30 km. The Faro and Atoll with the largest perimeter are both found in the Rhomboid Shoals in the interior of the Belize lagoon (S1 File).

Of the primary reef types, fringing and barriers constitute 98.9% of all reef types by number and 96% by length (Table 1). Fringing reefs alone compose 88.6% of the total number and 79.5% total length, with a mean length of 2 km per reef. By contrast, barrier reefs compose only 10.3% of the total number and 16.5% of the total length, with a mean length of ~3.5 km. Faros and atolls form only 1.1% by number and 4% by length, but have larger mean lengths (5.25 and 19.30 km respectively).

**Table 1. Number and length (km) of linear breakwater reefs (>500 m) and their subtypes in the Greater Caribbean.** Only 16 out of the possible 32 reef subtypes are present and, of those, 9 are common (bold) and 7 are uncommon (kmz data for all reefs provided in the S1 File).

| Total Number | 1023 | | | Total Length | 2,237 km | | |
|---|---|---|---|---|---|---|---|
| | | | | Mean | 2.2 km | | |
| Reef type | Fringing | Barrier | | Faro | Atoll | | |
| Total number (%) | 906 | 105 | | 10 | 2 | | |
| | (88.6) | (10.3) | | (1) | (0.2) | | |
| Total length, km (%) | 1,778 | 368 | | 53 | 39 | | |
| | (79.5) | (16.5) | | (2.3) | (1.7) | | |
| Mean length, km | 1.96 | 3.50 | | 5.25 | 19.30 | | |
| Reef subtype | TN | TL* | ML | Reef sub-type | TN | TL* | ML |
| 1. Interior FR flat type A* | 0 | 0 | 0 | 17. Coastal Faro flat type | 0 | 0 | 0 |
| 2. Interior FR crest type A | 0 | 0 | 0 | 18. Coastal Faro crest type | 0 | 0 | 0 |
| 3. Interior FR flat type D* | 1 | 2 | 1.82 | **19. Bank FR flat type** | **29** | **55** | **1.90** |
| 4. Interior FR crest type D | 1 | 1 | 0.51 | **20. Bank FR crest type** | **119** | **373** | **3.13** |
| 5. Interior BR flat type | 0 | 0 | 0 | 21. Bank BR flat type | 6 | 13 | 2.10 |
| 6. Interior BR crest type | 0 | 0 | 0 | **22. Bank BR crest type** | **39** | **262** | **6.72** |
| 7. Interior Faro flat type | 0 | 0 | 0 | 23. Bank Faro flat type | 0 | 0 | 0 |
| 8. Interior Faro crest type | 4 | 35 | 8.80 | 24. Bank Faro crest type | 6 | 17 | 2.88 |
| 9. Interior Atoll flat type | 0 | 0 | 0 | 25. Bank Atoll flat type | 0 | 0 | 0 |
| 10. Interior Atoll crest type | 1 | 23 | 23.02 | 26. Bank Atoll crest type | 1 | 16 | 15.57 |
| **11. Coastal FR flat type A** | **210** | **270** | **1.28** | 27. Oceanic BR flat type | 0 | 0 | 0 |
| **12. Coastal FR crest type A** | **18** | **37** | **2.07** | 28. Oceanic BR crest type | 0 | 0 | 0 |
| **13. Coastal FR flat type D** | **192** | **320** | **1.67** | 29. Oceanic Faro flat type | 0 | 0 | 0 |
| **14. Coastal FR crest type D** | **336** | **721** | **2.15** | 30. Oceanic Faro crest type | 0 | 0 | 0 |
| **15. Coastal BR flat type** | **36** | **48** | **1.33** | 31. Oceanic Atoll flat type | 0 | 0 | 0 |
| **16. Coastal BR crest type** | **24** | **45** | **1.89** | 32. Oceanic Atoll crest type | 0 | 0 | 0 |

*Total length, rounded to the nearest km. *A and D are attached and detached respectively.

Table 1 shows all possible reef subtypes (numbered 1–32), of which 16 are absent, including all six types of oceanic reef (27–32), six types of Interior reef (1,2,5,6,7,9), two types of coastal faros, and a single type of bank faro and atoll (Table 1). Several reef subtypes are also uncommon, including the remaining four interior reef types (3,4,8,10), and three bank types, including flat-type barriers and crest-type faros and atolls (21, 24, 26). That leaves only 9 common types (11–16,19,20,22), which consist of six types of coastal reef and three types of bank reef (Table 1).

Of the 16 subtypes present, eight are fringing reefs (3,4,11–14,19,20), two of which occur in interior settings, four in coastal settings, and two in bank settings. Of these eight, 52% are crest-type fringing reefs (n = 464) and 48% are flat-type fringing reefs (n = 429). There are four types of barrier reef present (15,16,21,22), two of which occur in coastal settings, and two in bank settings. Of these four barrier types, 60% are crest-types (n = 63) and the remaining 40% are flat-types (n = 42). The remaining four subtypes consist of two faro (8,24) and two atoll subtypes (10,26).

## Reef distribution

The common BW reef types are not distributed uniformly across the province. To explore these differences we subdivide the region into 8 marine ecoregions following Spalding et al [33]. In each, we plot the number of reef types and their lengths, and calculate their abundance by comparing the total length of reefs to the total length of coastline (Fig 3).

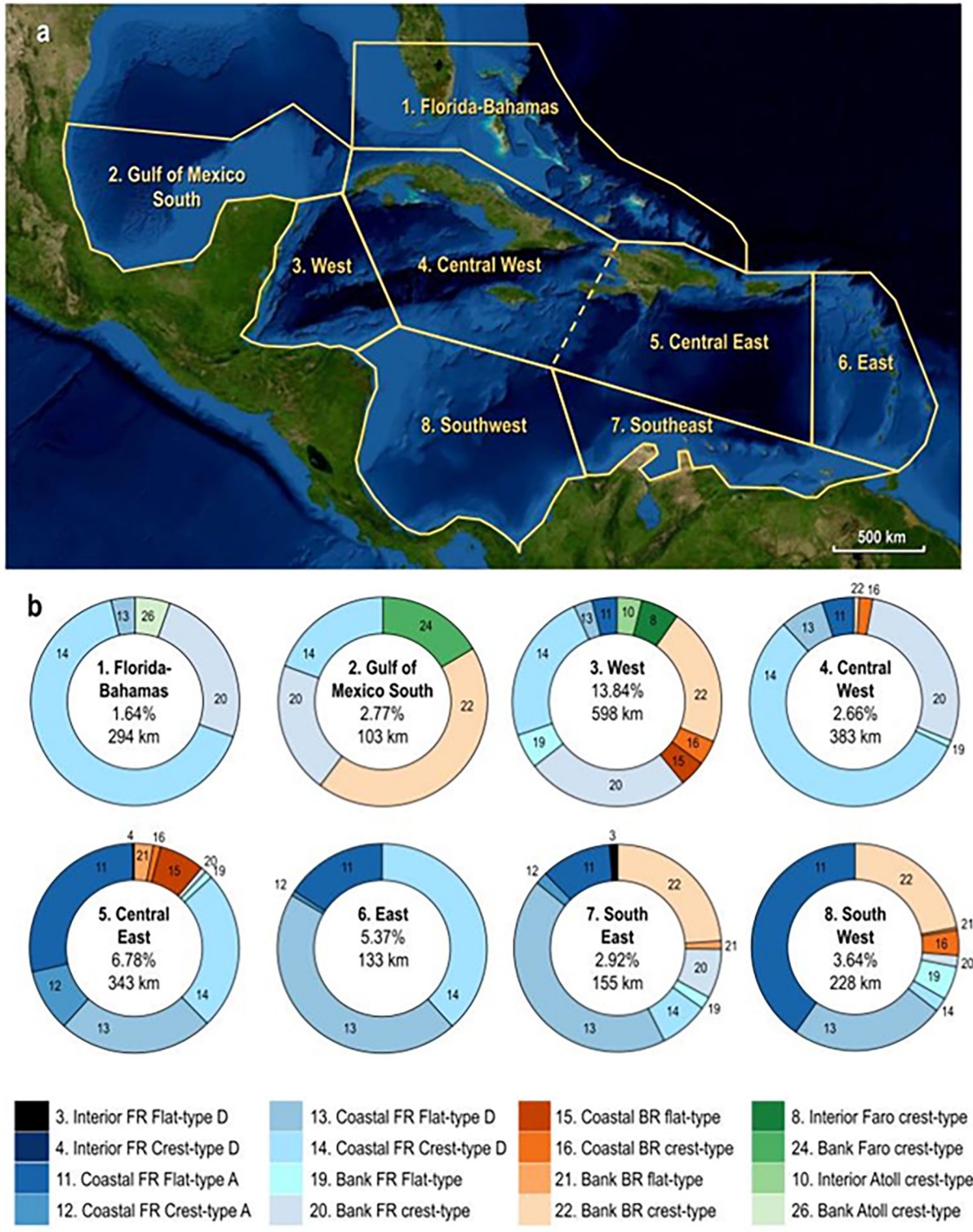

**Fig 3. Distribution of breakwater reef subtypes in 8 ecoregions of the Greater Caribbean.** a. Ecoregions, including the Gulf of Mexico South, Florida-Bahamas, and the West, East, Central (divided into Central East and Central West), Southeast, and Southwest Caribbean (modified after Spalding et al. 2007). b. Reef subtypes in each ecoregion, showing the total length of breakwater reefs (km), and their relative abundance considering the total coastal length (%). Note a bipartite distribution, with flat-type coastal reefs (11 and 13) dominating the Central East, East,

Southeast and Southwest Caribbean ecoregions, and crest-type reefs (14, 20, 22) dominating the Greater North, West, and Central-West Caribbean. Basemap data: NASA Visible Earth.

These data show significant heterogeneity in the abundance and distribution of reef subtypes both within and between ecoregions. The highest abundance of reefs occurs in the West Caribbean ecoregion where there are almost 600 km of BW reefs by length, which protect ~14% of the coastline. This ecoregion is also the only one where all of the common reef subtypes are present. The concentration of reefs increases at more local scales, and they protect as much as ~40% of coastal areas in small island groups like the Cayman and Bay Islands, and as much as 20–30% in small countries like Belize and central Panama. By contrast the Florida-Bahamas ecoregion has the lowest BW reef concentrations, and <2% of coastal areas are protected.

Significant disparity also occurs in the distribution of reef types, particularly crest vs. flat subtypes (Fig 3). Coastal flat-type reefs (11, 13, 15) form ~60% of all reef subtypes in the southern and eastern ecoregions (East 61%, Central East 59%, Southeast 54% and Southwest 65%). Whereas crest-type reefs (14, 20, 22) form ~80% of all subtypes in the northern and western ecoregions (Gulf of Mexico South 83.1%, Florida-Bahamas 90.8%, Central West 85.1%, West 70.2%). In addition, being less numerous, barrier reefs occur in clusters in several ecoregions, including the West (21%), Southwest (22%), Southeast (24%), and in the Gulf-of Mexico South (43%), where they dominate. This latter ecoregion is also the only one to have open-water bank faros (in contrast to those in interior protected settings).

## Reef morphology

Of the total number of reefs, 99% (96% by length) are either fringing or barrier reefs, which limits the reliability of morphological analysis of uncommon subtypes, such as faros or atolls. Excluding these, there are ten subtypes of fringing and barrier reefs. However flat-type bank barriers (21) are uncommon, leaving only nine common subtypes. Of these nine, two (12, 19) are not sampled in sufficient numbers to analyse statistically due to lack of clear imagery. As a consequence we analyse the remaining seven types in bank and coastal settings. Of these types, four are fringing (11, 13, 14, 20) and three are barriers (15, 16, 22), and three are flat-types from coastal settings (11, 13, 15), and four are crest-types from both coastal (14, 16) and bank (20, 22) settings (for examples of the common subtypes see video linked in S1 File).

These seven types of fringing and barrier reefs have discernible morphological differences, especially flat vs. crest types. Flat-types are more sinuous, especially when they are attached to the coast, and tend to have narrow, steeply sloping reef fronts, whereas crest-type reefs tend to be straighter and have wider reef fronts with more gentle slopes. To quantify these differences, we examine independent morphometric data collected from each of the seven subtypes, and use these data to examine the extent of variation within and between subtypes.

**Morphometric variation.** Geomorphic variation within the common reef subtypes is based on the statistical analysis of morphometric parameters from 179 linear reefs ($\geq$ 1000 m long), which provides an ~18% subset of the total number of linear BW reefs (Table 2). Of the ten metrics, covariates used to calculate morphometric values are excluded, leaving six that best represent the reef morphology (Table 2).

Internal variation within reef subtypes is analysed by calculating averages of each morphometric parameter, and their corresponding multivariate variance expressed as average deviations from the centroid; the complete data distribution is visualised using kernel density estimates (Fig 4). This analysis shows that, of the seven reef subtypes with sufficient data, four have narrow frequency distributions, implying an internally consistent morphology (Table 3, Fig 4). These include the three flat subtypes (11, 13, 15) and a crest-subtype (22), which have

**Table 2. Mean morphometric parameters of common reef subtypes.** Internal morphological variation within each subtype is represented by the average deviation from the group centroid. Many morphometric variables have non-normal sampling distributions causing the standard deviation to be larger than the mean of the variable.

| Reef Subtype | N | % CL Gaps ± SD | % CL Sinuosity ± SD | RF Av. Width (m) ±SD | BR Av. Width (m) ±SD | RF:BR Ratio ± SD | CL-MSB Distance (m) ± SD | Av. Cent. Deviation ± SE |
|---|---|---|---|---|---|---|---|---|
| 19. Bank FR Flat-type | 2 | 7.62 | 1.44 | 143.50 | 147.50 | 0.97 | 340.00 | NA* |
| | | ± 3.11 | ± 1.14 | ± 13.44 | ± 12.02 | ± 0.01 | ± 48.08 | |
| 15. Coast BR Flat-type | 7 | 4.03 | 10.18 | 82.12 | 143.37 | 0.61 | 229.38 | 2.05 |
| | | ± 3.92 | ± 11.09 | ± 41.74 | ± 58.74 | ± 0.32 | ± 144.63 | ± 0.39 |
| 22. Bank BR Crest-type | 13 | 2.32 | 6.61 | 103.20 | 129.13 | 0.99 | 407.09 | 2.59 |
| | | ± 3.06 | ± 7 | ± 62.66 | ± 84.96 | ± 0.71 | ± 304 | ± 0.24 |
| 11. Coast FR Flat-type A | 24 | 5.30 | 10.88 | 75.32 | 129.90 | 0.68 | 212.22 | 2.27 |
| | | ± 7.19 | ± 8.42 | ± 58.16 | ± 78.85 | ± 0.64 | ± 280.13 | ± 0.12 |
| 13. Coast FR Flat-type D | 36 | 4.57 | 10.21 | 79.85 | 147.98 | 0.58 | 248.90 | 2.54 |
| | | ± 4.45 | ± 11.98 | ± 39.66 | ± 60.40 | ± 0.29 | ± 158.26 | ± 0.16 |
| 12. Coast FR Crest-type A | 2 | 11.04 | 6.31 | 80.50 | 41.50 | 2.33 | 107.50 | NA* |
| | | ± 0.32 | ± 3.71 | ± 51.62 | ± 37.48 | ± 0.86 | ± 38.89 | |
| 14. Coast FR Crest-type D | 69 | 6.15 | 8.15 | 86.38 | 119.87 | 0.89 | 330.55 | 3.07 |
| | | ± 6.88 | ± 8.99 | ± 47.22 | ± 63.13 | ± 0.71 | ± 245.24 | ± 0.17 |
| 20. Bank FR Crest-type | 20 | 6.06 | 7.48 | 98.16 ± 58.1 | 132.98 | 0.89 ± 0.64 | 367.49 | 3.81 |
| | | ± 7.20 | ± 8.40 | | ± 78.84 | | ± 284.43 | ± 0.36 |
| 16. Coast BR Crest-type | 6 | 7.20 | 5.46 | 95.33 | 107.56 | 1.10 | 436.50 | 4.06 |
| | | ± 7.18 | ± 8.42 | ± 58.16 | ± 78.85 | ± 0.65 | ± 435.64 | ± 0.42 |

\* Insufficient data.

relatively low average deviations from the centroid (~2–2.5; Fig 5). The three remaining crest subtypes (14, 16, 20) have higher deviations from the centroid (~3–4) and multimodal distributions, implying more internal morphometric variation and less consistent morphologies (Figs 4 and 5).

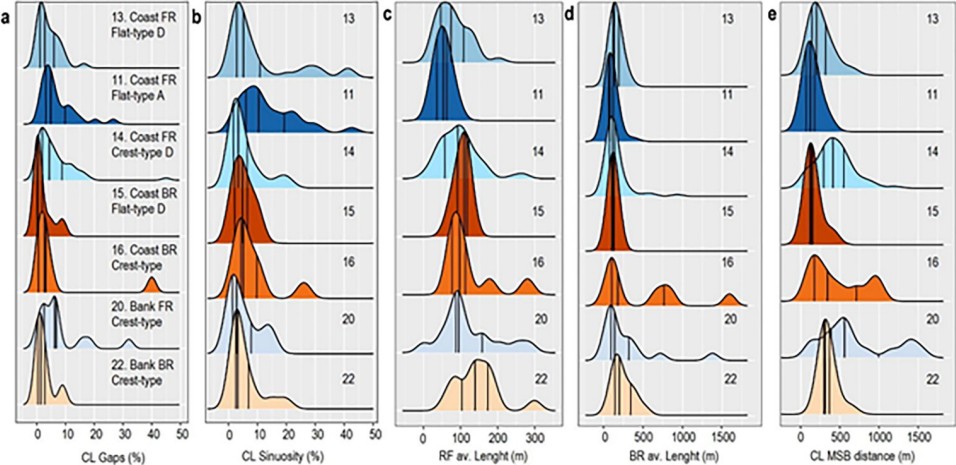

**Fig 4. Kernel density estimates of morphometric data described in Table 3 with quartiles.** Panels a and b show density curves corresponding to crestline sinuosity and persistence (% gaps) by reef subtype (in percentages). Panels c, d and e show density curves in reef-front and back-reef areas (av. length) and reef-front slope (Crestline-MSB distance). Of those with consistent morphologies (11, 13, 15, 22) all three flat types (11, 13, 15) show back-reef areas with the same normal distribution, central tendency and narrow dispersion, and steep reef-front slopes.

**Table 3. Relative contribution of morphometric parameters to common reef subtypes.**

| Reef Subtype | Av. Sq. Distance | Morphometric Variable | Av. Value | Av.Sq. Dist. | Sq.Dist SD | Contrib % | Cum. % |
|---|---|---|---|---|---|---|---|
| **15.** Coastal Barrier Reef Flat Type | 2.25 | BR av. length (m) | -0.24 | 0.04 | 0.54 | 1.79 | 1.79 |
| | | RF av. length (m) | 0.30 | 0.07 | 0.54 | 3.04 | 4.83 |
| | | CL Sinuosity (%) | -0.35 | 0.15 | 0.55 | 6.70 | 11.53 |
| | | CL MSB distance (m) | -0.48 | 0.45 | 0.46 | 20.07 | 31.60 |
| | | CL gaps (%)* | -0.73 | 1.54 | 0.58 | 68.40 | 100.00 |
| **11.** Coastal Fringing Reef Flat Type A | 2.2 | BR av. length (m) | -0.39 | 0.12 | 0.29 | 5.33 | 5.33 |
| | | RF av. length (m) | -0.74 | 0.14 | 0.51 | 6.40 | 11.74 |
| | | CL MSB distance (m) | -0.74 | 0.16 | 0.44 | 7.09 | 18.83 |
| | | CL gaps (%)* | 0.48 | 0.35 | 0.45 | 16.07 | 34.90 |
| | | CL Sinuosity (%) | 0.65 | 1.43 | 0.43 | 65.10 | 100.00 |
| **13.** Coastal Fringing Reef Flat Type D | 4.67 | BR av. length (m) | -0.11 | 0.11 | 0.45 | 2.34 | 2.34 |
| | | CL MSB distance (m) | -0.38 | 0.35 | 0.43 | 7.42 | 9.76 |
| | | RF av. length (m) | -0.19 | 1.05 | 0.33 | 22.41 | 32.18 |
| | | CL gaps (%)* | -0.40 | 1.12 | 0.48 | 24.02 | 56.19 |
| | | CL Sinuosity (%) | 0.44 | 2.05 | 0.44 | 43.81 | 100.00 |
| **14.** Coastal Fringing Reef Crest Type D | 3.42 | CL Sinuosity (%) | -0.29 | 0.41 | 0.38 | 11.92 | 11.92 |
| | | BR av. length (m) | -0.10 | 0.50 | 0.24 | 14.61 | 26.53 |
| | | RF av. length (m) | 0.03 | 0.82 | 0.37 | 23.84 | 50.38 |
| | | CL gaps (%)* | 0.09 | 0.84 | 0.44 | 24.40 | 74.78 |
| | | CL MSB distance (m) | 0.33 | 0.86 | 0.42 | 25.22 | 100.00 |
| **20.** Bank Fringing Reef Crest Type | 9.4 | CL Sinuosity (%) | -0.31 | 0.38 | 0.50 | 4.02 | 4.02 |
| | | CL gaps (%)* | 0.29 | 1.29 | 0.47 | 13.73 | 17.75 |
| | | RF av. length (m) | 0.43 | 1.88 | 0.46 | 19.99 | 37.74 |
| | | CL MSB distance (m) | 1.01 | 2.72 | 0.47 | 28.94 | 66.69 |
| | | BR av. length (m) | 0.45 | 3.13 | 0.35 | 33.31 | 100.00 |
| **16.** Coastal Barrier Reef Crest Type | 11.53 | CL Sinuosity (%) | 0.01 | 0.77 | 0.40 | 6.72 | 6.72 |
| | | CL gaps (%)* | -0.17 | 1.08 | 0.47 | 9.40 | 16.11 |
| | | RF av. length (m) | 0.46 | 1.47 | 0.42 | 12.76 | 28.87 |
| | | CL MSB distance (m) | 0.39 | 1.59 | 0.55 | 13.79 | 42.66 |
| | | BR av. length (m) | 1.51 | 6.61 | 0.48 | 57.34 | 100.00 |
| **22.** Bank Barrier Reef Crest Type | 3.14 | CL MSB distance (m) | -0.20 | 0.11 | 0.41 | 3.62 | 3.62 |
| | | CL Sinuosity (%) | -0.29 | 0.28 | 0.47 | 8.77 | 12.39 |
| | | BR av. length (m) | 0.15 | 0.43 | 0.45 | 13.69 | 26.08 |
| | | RF av. length (m) | 0.72 | 1.11 | 0.48 | 35.26 | 61.34 |
| | | CL gaps (%)* | -0.39 | 1.21 | 0.52 | 38.66 | 100.00 |

* Logarithmically transformed variable.

Given the presence of multimodal data, a multivariate analysis of dispersion and central tendency is used to compare the degree of internal morphometric variation among each subtype and visualised through a boxplot and an mMDS of bootstrapped-region estimates of data means (Fig 5). These plots corroborate that flat types are morphologically more consistent, with less internal variation than crest-subtypes (Fig 5A). This is also shown by the multivariate central tendency shown in Fig 5B, where coastal flat subtypes (11, 13, 15) have lower uncertainty in central tendencies, and bank and coastal crest subtypes (16, 20, 22) have greater uncertainty.

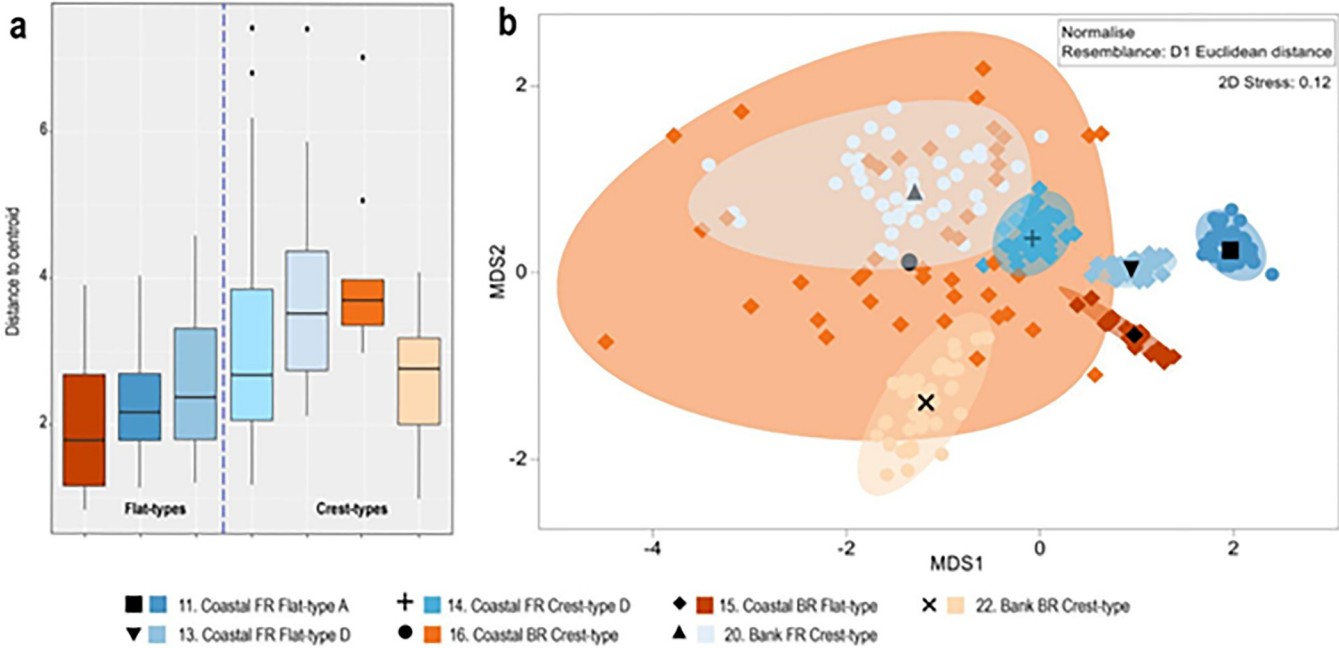

**Fig 5. Multivariate analysis of dispersion and central tendencies of morphometrics in the seven common reef subtypes.** a: Boxplot of multivariate dispersion (variance), showing the median and interquartile ranges of distances to the group centroid. Note: dashed line divides flat and crest subtypes, and crest types have outliers in 3 of the 4 groups. b. Multivariate Central Tendency of morphometrics. The mMDS plot shows bootstraps averages together with 95% region estimates of uncertainty for 'mean settings' (black symbols) of each reef subtype. Bootstrapping approximation is obtained from the distances in an m = 5-dimensional mMDS space, for which the Pearson correlation to the dissimilarities is = 0.993. The overlapping diffuse subtypes (14, 16, 20, 22), with a high average squared Euclidean distance, indicate higher uncertainty in the central tendency (implying indistinct morphologies) compared to the 3 other subtypes (11, 13, 15) where the uncertainty in central tendency is considerably lower (implying distinct morphologies).

**Morphometric differences.** Morphological differences between reef subtypes are identified by the average morphometric parameters in Table 2. For example, both coastal fringing and barrier flat-subtypes (11, 15) have more persistent crestlines (< 5% gaps), are moderately sinuous (~10%), with large back-reef flats (130–140 m), and narrow (~75- 80m), steeply sloping reef-fronts (230-250m). Crest subtypes (22), by contrast have less sinuous crestlines (7%), roughly equal-sized back-reef and reef-front areas, and more gentle reef-front slopes. The other crest subtypes (14, 16, 20), although having less consistent morphologies, have similarly persistent crestlines (6–7%), low sinuosities (6–8%), equal-sized back-reef and reef-front areas (100–130 m), and wider (85–100 m), gentler reef front slopes (330-430m).

These differences are corroborated by calculating the relative contribution of individual morphometric variables to each subtype, using a similarities percentage breakdown for the repeated average values of each variable, and ordering its contribution according to the most consistent value (Table 3). This analysis confirms that flat subtypes (11, 13, 15) are primarily characterised by the average size of back-reef and reef-front zones, which vary by only ~3 and 5% respectively. By contrast, Crest subtypes (14, 16, 20), are better characterised by crestline sinuosity and persistence, which vary by ~8 and 12% respectively.

Differences between the seven reef subtypes are further assessed by a comparative hierarchical analysis of the three main classification levels (reef type, setting and shallow geomorphology) shown in S1 File. At the shallow geomorphology level, 78% of flat or crest-subtypes have morphometrics that are consistent with the group. This decreases slightly in the other levels (setting and reef type) where 75% of morphometrics are consistent. However, when morphometric data are analysed according to all three classification levels, only 39% of data is

consistent with a particular subtype. This implies all three levels are required for optimal classification.

## Discussion

The *a priori* geomorphic classification of linear BW reefs employed here is based on a standardisation of environmental and morphological parameters that have historically been considered important in reef characterization, such as form, depth, physiographic setting, and geomorphic zonation [1, 33–36]. As such it has several advantages: first, it facilitates the accurate communication of reef character between scientific disciplines. Second, it provides a framework which can be expanded to include dispersed and submerged reef classes. And finally, it allows the testing of hypotheses on reefs between and within lagoons, shelves, banks and oceans, where potential environmental controls, such as bathymetry, slope, tides and wave fields, differ markedly. In addition to these advantages, the classification is easy to apply given that it only requires two data sources: clear aerial or high-resolution satellite images (available in virtual globes like Google Earth) and metre-scale bathymetry (available in navigation charts). However, there are some inherent limitations in the available public-domain imagery. Google Earth, for example, is biassed towards populated areas leaving remote parts of the Caribbean with either limited or lower-resolution data. As a result there is bias in classifying and collecting reef data, particularly in non-coastal settings such as banks and open-sea areas.

Notwithstanding this bias, our abundance data clearly shows that 80% of linear BW reefs in the Greater Caribbean are fringing reefs by length, and that these consist of six common subtypes in both bank and coastal settings. Barrier reefs are less common at 16% by length and have three common subtypes in bank and coastal settings. Faros and atolls are rare at 4% by length and only occur in bank and interior settings. Previous surveys of Caribbean reef types have found similar proportions, but the categories were based on different criteria. For example Milliman [9] suggested that there were only two true barriers (Belize and Providencia) but noted several intermediate 'fringing-barrier' types with 10 m deep lagoons. By standardising barriers as those with lagoons 10 m or deeper, we count more than 100 barriers in the province. Similarly, Milliman suggested that there were ten atolls with topographically 'enclosed lagoons'. We count a similar number but restrict the atoll category to landless lagoons enclosed 75% or more by linear BW reefs. This stricter definition of atolls emphasises enclosure by BW reefs and discounts lagoons enclosed by other sedimentary deposits such as sand shoals or submerged or nonlinear reefs, thereby facilitating comparison with atolls elsewhere [37]. It also means that many of Milliman's 'enclosed atolls' are counted as barrier reefs in our standardised classification because their BW reefs encircle less than 75% of the lagoon.

In addition to the abundance of reef categories, the independent morphometric data allows us to explore differences within and between reef subtypes at the classification level. However, due to the limited sample size imposed by the lack of clear imagery in Google Earth, these data are insufficient to determine the full extent of differences between all subtypes in the classification. Nevertheless, of the seven subtypes with sufficient data, there are clear and consistent differences at the shallow-geomorphology level between flat-subtypes and crest-subtypes. Flat-subtypes, with mostly unimodal morphometrics, have the most consistent morphologies and typically have sinuous crestlines, large back-reef zones and comparatively smaller, steeper, reef-front zones. By contrast, crest-subtypes with multimodal morphometrics, have less-consistent morphologies, but tend to be straighter and have more equal-sized back-reef and reef-front zones. These differences between flat- and crest-subtypes imply that there are at least two reef morphotypes in the Caribbean. The multimodal character of crest-type reefs, however, implies there are unresolved spatio-temporal variants at either the classification level, or due to variation of other environmental parameters.

The distinct differences between these two morphotypes are also supported by their different distributions, with flat-subtypes dominating in southern ecoregions and crest-subtypes dominating in northern ecoregions. These differences stem largely from the relative rarity of flat-subtypes in the north, with ecoregions such as the Bahamas and Florida having the least (9%), gradually increasing in central ecoregions (15–30%), and eventually co-dominating with crest-type reefs in the south (55–65%). This northerly decrease therefore accounts for the differences between reefs that caused so much confusion during the early descriptions of Caribbean reef ecology and morphology [2, 4, 6, 9]. Northerly reefs lack algal ridges and reef flats not because they are in an immature stage of development, but because they are different reef morphotypes than the flat-subtypes found further south.

Taken together, the differences in morphometrics and distribution imply that flat- and crest-subtypes are unlikely to be stages in the development of a single reef type, but instead represent fundamentally different reef morphotypes. This is corroborated by reports of differences in their internal structure. Early Caribbean reef models were based on a drill transect from a flat-type coastal fringing reef at Galeta Point, Panama [12]. This 12-core transect revealed a 15 m thick reef-crest section dominated by a framework of *Acropora palmata*, which had accreted vertically during the last 8 ka keeping pace with late Holocene SL rise. By contrast, a 3-core drill transect from a crest-type bank fringing reef at Long Reef in the Florida Keys showed a 1 to 5 m thick cap of *A. palmata* underlain by loose sand and coral heads [15]. A similar structure was also reported from a 12-core transect on a crest-type coastal fringing reef at Punta Maroma in the Yucatan, Mexico [38]. In this case, a 2-m thick layer of *A. palmata* gravel capped a thicker sand section below, and dating showed the thin, gravel-dominated crest retrograding over the back reef in the last 6 ka. The retrogressive detrital model of crest-type reefs from the Yucatan thus differs significantly from the aggradational framework model of the flat-type reef at Panama.

Having not included environmental data in the morphological analysis, we cannot draw conclusions about what controls the differences between crest- and flat-morphotypes. But the drilling reconstructions do offer some insight. The internal structure of crest-type reefs for example is postulated to be controlled by the cyclic destruction and regeneration of reef-crest corals due to repeated hurricane impacts over thousands of years [38]. The predominance of crest-type reefs in northern ecoregions may therefore reflect the northerly increase in hurricane frequency (e.g. [39, 40]). What is unclear however is why flat-type reefs are less common in these areas, given that they co-occur with crest type reefs in other ecoregions. In regions where hurricanes have low frequencies, such as Panama, flat-type fringing reefs develop Indo-Pacific style algal ridges and extensive reef flats which are reliant on the binding role of crustose coralline algae (e.g. [41]). The prevalence of reef-building corallines in these southern flat-type reefs may be related to a stronger wave climate, which is intensified by the Caribbean Low-Level Jet [16, 42, 43]. Alternatively, it may be related to the latitudinal decrease in annual temperature range, which was initially speculated to be the cause of limited coralline abundance in more northerly reefs [8].

If the development of crest and flat morphotypes is indeed controlled by different environmental parameters, then this predicts significant differences in their ecologies. Crest-types adapted to frequent hurricanes, for example, may have a lower diversity due to frequent disturbance [44], and more inter-reef variability in coral assemblages due to local variations in hurricane impact [45–48]. By contrast, the less impacted flat types may have assemblages that are more stable and correspond to wave-energy variation [14, 18].

In conclusion, our standardisation of reef categories provides a widely applicable and objective framework within which to compare their morphometric variation and assess the spatio-temporal controls on their distribution and development. Applying this classification to the

greater Caribbean shows that there are over 2000 km of linear breakwater reefs, which fall into 16 reef categories, 9 of which are common. Barrier reefs, faros and atolls are relatively uncommon, accounting for only 20% of the total length, which implies that additional processes and/ or specific settings are required for their development compared to fringing reefs. Bank and coastal Faros, for example, are only found in the southern Gulf of Mexico which, unlike other areas, has a seasonal multidirectional wave field [43].

Fringing reefs are by far the most common type, comprising 80% of the total BW reef length. Like the other types, they have at least two distinct morphotypes which have different mechanisms of accretion and development. Crest-types dominate in areas exposed to hurricanes, and have detrital internal structures which demonstrate cyclic accretion due to repeated skeletal destruction and deposition through time [38]. Their multimodal morphometrics, however, imply the presence of additional unresolved morphotypes within the group. By contrast, flat-types have a more distinct morphology, increase in abundance to the south, and have internal structures that reflect the vertical accretion of reef-crest framework through time, implying they are relatively undisturbed by hurricanes and more reliant on ecological differences.

Although not investigating the specific cause of the geomorphic and distributional differences between crest and flat type reefs, we have clearly shown that they are distinct reefal geoforms and thus unlikely to be successional stages of a single reef type, nor a genetically related sequence of types. If further work can discover the cause of these differences, it will have significant implications not only for how reefs are studied, assessed and managed, but also how they might respond to anthropogenic-induced changes in climate and sea level.

## Supporting information

**S1 File. KMZ files, reef length table, classification cross-validation figure, reef zone definitions, and YouTube video link.**
(DOCX)

## Acknowledgments

We thank Alexandra Elbakyan (Kazakhstan) and Julián Quintero-Ibáñez (Colombia) for providing access to data and literature, Dolores Ruiz-Gonzalez (Mexico) for help with the paper's YouTube video (https://youtu.be/dIr67dFwtO4), and the 2 anonymous journal reviewers for their constructive comments.

## Author Contributions

**Conceptualization:** Paul Blanchon.

**Data curation:** Paul Blanchon, Alexis E. Medina-Valmaseda, Eduardo Islas-Domínguez, Edlin Guerra-Castro, Adan Guillermo Jordan-Garza, Paula A. Zapata-Ramírez.

**Formal analysis:** Paul Blanchon, Alexis E. Medina-Valmaseda, Eduardo Islas-Domínguez, Edlin Guerra-Castro.

**Funding acquisition:** Paul Blanchon, Alexis E. Medina-Valmaseda.

**Investigation:** Paul Blanchon, Alexis E. Medina-Valmaseda, Edlin Guerra-Castro.

**Methodology:** Paul Blanchon, Alexis E. Medina-Valmaseda, Edlin Guerra-Castro.

**Project administration:** Alexis E. Medina-Valmaseda.

**Software:** Edlin Guerra-Castro.

**Writing – original draft:** Paul Blanchon, Alexis E. Medina-Valmaseda, David Blakeway.

**Writing – review & editing:** Paul Blanchon, Alexis E. Medina-Valmaseda, Eduardo Islas-Domínguez, David Blakeway, Joaquín Rodrigo Garza Pérez, Adan Guillermo Jordan-Garza, Ismael Mariño-Tapia, Paula A. Zapata-Ramírez.

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
