## [Decision Letter · Decision Letter 0]

17 May 2022

PONE-D-22-06844Linear Breakwater Reefs of the Greater Caribbean: Classification, Distribution & Morphology.PLOS ONE

Dear Dr. Blanchon,

Thank you for submitting your manuscript to PLOS ONE. After careful consideration, we feel that it has merit but does not fully meet PLOS ONE’s publication criteria as it currently stands. Therefore, we invite you to submit a revised version of the manuscript that addresses the points raised during the review process.

Dear Paul, I am finally able to send you the comments on your manuscript. The two reviewers greatly appreciated your manuscript and they suggest only a few minor revisions.I agree with them and I greatly appreciate your approach in classifying reefs. I think we will proceed with a snappy acceptance soon after minor changes will be made.==============================

We look forward to receiving your revised manuscript.

Kind regards,

Andrea Zerboni, Ph.D.

Academic Editor

PLOS ONE

Journal Requirements:

“This work is supported by SEP-CONACyT grant A1-S-18879 and PAPIIT grant IN214819 to PB. We thank the 2 anonymous journal reviewers for their constructive comments.”

“This work is supported by SEP-CONACyT grant A1-S-18879 and PAPIIT grant IN214819 to PB.”

“This work is supported by SEP-CONACyT grant A1-S-18879 and PAPIIT grant IN214819 to PB.”       

5. We note that Figure 3 in your submission contain [map/satellite] images which may be copyrighted. All PLOS content is published under the Creative Commons Attribution License (CC BY 4.0), which means that the manuscript, images, and Supporting Information files will be freely available online, and any third party is permitted to access, download, copy, distribute, and use these materials in any way, even commercially, with proper attribution. For these reasons, we cannot publish previously copyrighted maps or satellite images created using proprietary data, such as Google software (Google Maps, Street View, and Earth). For more information, see our copyright guidelines: http://journals.plos.org/plosone/s/licenses-and-copyright.

   a. You may seek permission from the original copyright holder of Figure 3 to publish the content specifically under the CC BY 4.0 license. 

6. We note that Figure 4 in your submission contain copyrighted images. All PLOS content is published under the Creative Commons Attribution License (CC BY 4.0), which means that the manuscript, images, and Supporting Information files will be freely available online, and any third party is permitted to access, download, copy, distribute, and use these materials in any way, even commercially, with proper attribution. For more information, see our copyright guidelines: http://journals.plos.org/plosone/s/licenses-and-copyright.

    1. You may seek permission from the original copyright holder of Figure 4 to publish the content specifically under the CC BY 4.0 license.

Reviewers' comments:

Reviewer's Responses to Questions

**Comments to the Author**

1. Is the manuscript technically sound, and do the data support the conclusions?

Reviewer #1: Yes

Reviewer #2: Yes

2. Has the statistical analysis been performed appropriately and rigorously? 

Reviewer #1: Yes

Reviewer #2: Yes

3. Have the authors made all data underlying the findings in their manuscript fully available?

Reviewer #1: Yes

Reviewer #2: Yes

4. Is the manuscript presented in an intelligible fashion and written in standard English?

Reviewer #1: Yes

Reviewer #2: Yes

5. Review Comments to the Author

Reviewer #1: There is an adage that most models are either too good to be true or too true to be good. The former are so general that their descriptive and predictive powers are weak, whereas the latter are so intricate and detailed that each case is its own, ungeneralizable category. In this paper, Blanchon et al. provide a hierarchical classification of coral reefs of the Caribbean. Using Google Earth and bathymetric charts, they quantify the occurrence of the reef types by ecoregion and make a first cut at analyzing the morphometric variability of the different types they recognize. Their classification is far more detailed and rigorous than the simple fringing (no lagoon), bank/barrier (narrow lagoon), barrier (broad lagoon), and atoll (or carbonate-bank; enclosed lagoon) categories commonly in use by ecologists. Their contribution is potentially game-changing because it could dispense with a lot of less-precise classifications that have been published, adding rigor to the study of reef geomorphology. It also got me thinking about the nexus of reef geology and reef ecology, per the next paragraph.

Is this more detailed classification indeed more useful, or is it too true to be good? Clearly, this work will be of value to the authors and other reef geologists interested in understanding the physical controls on reef development. The two paragraphs starting on Line 541 amply demonstrate why this work is interesting and important, and they lay the groundwork for future tests of important geological hypotheses. But what about those pesky ecologists, who will likely have a hard time learning all the categories—and learning to use them effectively? The authors would serve themselves well to expand the Discussion a little, adding a few sentences to explain that it might be possible to make ecological predictions as consequences of the physical drivers of reef development. Alternatively, maybe the phenomena arrayed on ecological hierarchies transcend or are not affected by these differences of geomorphological classification (see for example Murdoch and Aronson, 1999, Coral Reefs). It would be fascinating to test the limits of the hypothesis that physics controls (almost) everything, as I suspect it does; but is the control geomorphological or the parochial result of environmental idiosynchrasy? These kinds of questions make the paper important.

Specific Comments.

Comment 1: do not abbreviate sea level as SL, especially in the abstract. It is not a good look.

Comment 2: Geister (1977) is cited, appropriately, but Hubbard’s 3-D environmental model of reef development is missing. That model might or might not be a counterpoint to what Blanchon and colleagues are trying to get across in their paper. Either way, it should be cited if only for completeness. It is certainly germane to the paragraphs of the Discussion beginning on lines 530 and 557.

Comment 3: the word data is sometimes used as a singular. Please go through the manuscript and correct that.

Line 125: allow, not allows.

Line 164: faros, not faro’s

Fig. 2: This figure consists of two panels, but the caption appears only to describe the top panel. A few words about the bottom panel would be appropriate, especially because the view is orthogonal to the view in the top panel. Where is the reef in the photo?

Line 209: The definition of tract is at variance with its common use to describe the totality of reef development in southern Florida: the Florida reef tract. Is there an alternative word the authors can employ to avoid confusion?

Line 229: CL, BR, RF…this sentence sounds terrible when read aloud. Why not just spell out the terms to make it easy on the reader? That is what’s done in lines 231–232.

Line 382: The sentence beginning with Whereas is not a sentence. Combine it with the previous sentence using a comma.

Reviewer #2: A well-written and interesting paper with a logical initial approach to categorizing reef morphology using Google Earth that can be applied to regions outside the Caribbean. I find little to fault and have only three minor comments and list a few typo/formatting issues.

1. Strictly for reproducibility, if feasible indicate whether the bathymetry for each reef was sourced from the public domain or proxy.

2. Line 336, it would be useful to briefly summarize how the ecoregions were determined for those unfamiliar with Spalding et al.’s work – such as myself.

3. Line 346, I am not sold on “15 Coastal BR flat-type” reefs dominating the Central East, East, Southeast and Southwest Caribbean ecoregions. From a purely visual analysis of your distribution using Figure 3, types 14 and 22 have a much higher dominance than 15, which appears to be present only in the Central East at a lesser extent than type 14. Including type 15 needs a better argument or explanation.

• Line 149, check font size and line space formatting

• Line 170-171, font color. I think maybe your list numbers were classified as references.

• Line 204, should this be Supplementary Information SI-2?

• Line 227, check line space formatting

• Line 304, should this refer to Supplementary Table S1?

• Line 471, should this be supplementary figure S1?

6. PLOS authors have the option to publish the peer review history of their article (what does this mean?). If published, this will include your full peer review and any attached files.

Reviewer #1: No

Reviewer #2: No

---

## [Author Response · Author response to Decision Letter 0]

2 Jun 2022

1. Please ensure that your manuscript meets PLOS ONE's style requirements, including those for file naming. DONE

2. In your Data Availability statement, you have not specified where the minimal data set underlying the results described in your manuscript can be found. Minimal underlying data set is in the Supporting Information 

3. Thank you for stating the following in the Acknowledgments Section of your manuscript: . DONE

4. Thank you for stating the following financial disclosure: “This work is supported by SEP-CONACyT grant A1-S-18879 and PAPIIT grant IN214819 to PB.” The funders had no role in study design, data collection and analysis, decision to publish, or preparation of the manuscript."

5. We note that Figure 3 in your submission contain [map/satellite] images which may be copyrighted. We have removed the image and replaced it with a public domain version

6. We note that Figure 4 in your submission contain copyrighted images. We have removed the image.

Please review your reference list to ensure that it is complete and correct. DONE

---

## [Editor Report · Decision Letter 1]

3 Jun 2022

Linear Breakwater Reefs of the Greater Caribbean: Classification, Distribution & Morphology.

PONE-D-22-06844R1

Dear Dr. Blanchon,

We’re pleased to inform you that your manuscript has been judged scientifically suitable for publication and will be formally accepted for publication once it meets all outstanding technical requirements.

Kind regards,

Andrea Zerboni, Ph.D.

Academic Editor

PLOS ONE
---

## [Editor Report · Acceptance letter]

4 Aug 2022

PONE-D-22-06844R1 

Linear Breakwater Reefs of the Greater Caribbean: Classification, Distribution & Morphology. 

Dear Dr. Blanchon:

I'm pleased to inform you that your manuscript has been deemed suitable for publication in PLOS ONE. Congratulations! Your manuscript is now with our production department. 

Kind regards, 

on behalf of

Prof. Andrea Zerboni 

Academic Editor

PLOS ONE